# Regulation of locomotor speed and selection of active sets of neurons by V1 neurons

Yukiko Kimura [1,2] & Shin-ichi Higashijima [1,2]

During fast movements in vertebrates, slow motor units are thought to be deactivated due to the mechanical demands of muscle contraction, but the associated neuronal mechanisms for this are unknown. Here, we perform functional analyses of spinal V1 neurons by selectively killing them in larval zebrafish, revealing two functions of V1 neurons. The first is the long-proposed role of V1 neurons: they play an important role in shortening the cycle period during swimming by providing in-phase inhibition. The second is that V1 neurons play an important role in the selection of active sets of neurons. We show that strong inhibitory inputs coming from V1 neurons play a crucial role in suppressing the activities of slow-type V2a and motor neurons, and, consequently, of slow muscles during fast swimming. Our results thus highlight the critical role of spinal inhibitory neurons for silencing slow-component neurons during fast movements.

[1] National Institutes of Natural Sciences, Exploratory Research Center on Life and Living Systems (ExCELLS), National Institute for Basic Biology, Okazaki, Aichi 444-8787, Japan. [2] The Graduate University for Advanced Studies, Okazaki, Aichi 444-8787, Japan. Correspondence and requests for materials should be addressed to S.-i.H. (email: shigashi@nibb.ac.jp)

Vertebrates can produce movements of widely varying strength and speed by using rhythmic networks of neurons located in the spinal cord. A leading model for determining the recruitment patterns of motoneurons (MNs) during speed/strength changes is the size principle[1–6]. According to this principle, the pool of active cells steadily increases in size with progressive increases in the force and speed of movement. It is assumed that MNs that are bigger in size and innervate fast-type muscles are added to smaller MNs that innervate slow-type muscle as the speed/strength of movement increases.

Indeed, in larval zebrafish, it has been found that MNs of larger size are recruited only during strong/fast movements[7,8]. Similar phenomena were also found in a class of excitatory premotor interneurons (V2a neurons)[8,9]. Studies in larval zebrafish, however, have also shown that recruitment patterns did not perfectly follow a simple adding rule. Slow-type MNs as well as slow-type V2a neurons were found to actually be inactive during stronger/faster movements[9–12], suggesting that, with increasing speed/strength of movements, deactivation of slow-type MNs and interneurons occurs. Deactivation of slow-type muscles during very fast swimming or escape behavior was also reported in adult fish[13–15]. Unlike fish, skeletal muscles in mammals consist of mixed fibers with slow- and fast-type muscle fibers intermingled, which makes it more difficult to accurately examine the activities of slow- and fast-type muscles separately. Nonetheless, several lines of evidence suggest that orderly recruitment of motor units does not always occur in mammals[16], and preferential recruitment of faster muscle fibers during rapid contractions is suggested to occur to meet the mechanical demands of fast contraction and relaxation of muscles[17–21]. Importantly, however, the neuronal basis for the silencing of slow-component neurons during fast/strong movements has remained largely unknown in any vertebrate species.

In order to understand the neuronal circuits that control locomotion, it is important to identify the various types of neurons composing the spinal central pattern generators (CPGs). For the past 20 years, the leading strategy for the identification of neuronal types in the spinal cord has been to utilize various transcription factors that are expressed in a subset of neurons during development[22–25]. Importantly, the expression patterns of the transcription factors are mostly conserved across vertebrate species, making cross-species comparisons possible.

V1 neurons are one class of neurons that are defined by the expression of En1. V1 neurons are ipsilaterally projecting inhibitory neurons in vertebrates thus far examined[26–29]. In larval zebrafish and frog tadpoles, these neurons generally fire in phase with MNs located nearby during swimming and are proposed to provide in-phase inhibition to CPG and motor neurons to help terminate the firing of the target neurons in each cycle during swimming[27,28]. In this scheme, inactivation of V1 neurons would be expected to prolong firings of the CPG and motor neurons in each cycle and, consequently, prolong the cycle period. This kind of genetic inactivation of V1 neurons was performed in mice[30], which showed that the cycle period was indeed prolonged during locomotor-like activities in V1-deficient neonatal mice.

Here we performed functional analyses of V1 neurons in larval zebrafish by selectively killing spinal V1 neurons using diphtheria toxin A (En1-DTA fish). We have revealed two functions of V1 neurons. The first is the long proposed role of V1 neurons. In En1-DTA fish, the cycle period in swimming was prolonged. The second is completely new: V1 neurons were found to play an important role in the selection of active sets of neurons. In En1-DTA fish, slow-type V2a neurons and slow-type MNs were vigorously active during strong movements. We have thus succeeded in identifying the neuronal basis that accounts for the silencing of slow-component neurons during fast/strong movements.

## Results

**Firing patterns of V1 neurons during fictive swimming.** We performed loose-patch electrophysiological recordings of V1 neurons together with ventral root (VR) recordings during variable speeds of fictive swimming using 3-day post fertilization (dpf) larvae of Tg[*en1b*:Gal4; UAS:Kaede][31] (Fig. 1a, b). In typical swimming episodes elicited by brief electrical stimulation (ES; Fig. 1b), swimming speed was initially fast (fast swim in Fig. 1c, d). The fast swim is followed by slow swim, the frequency of which is around 25–35 Hz (slow swim in Fig. 1c, d).

The spiking patterns of V1 neurons differed from cell to cell but could be approximately categorized into two groups. In one group, spiking activity mainly occurred during the initial fast phase of the swimming episode (Fig. 1c). We call these cells fast-type V1. In the other group, spiking activity mainly occurred during the late phase of slow swimming (Fig. 1d). We call these cells slow-type V1.

For both types of V1 neurons, spiking activities generally occurred when nearby VR activities were high (Fig. 1c, d, right). We performed a more quantitative phase analysis (Supplementary Fig. 1). Briefly, the middle time point of one VR burst was set as time 0, and the next time point was set as time 1. Figure 1e shows the histograms of spike timings of fast-type V1 neurons during fast swim, whereas Fig. 1f shows the histograms of spike timings of slow-type V1 neurons during slow swim. In both cases, spiking activities are centered around time 0, indicating that the firing activities of V1 neurons occurred in phase with the nearby VR activities regardless of the swimming speed.

The subdivision of V1 neurons with respect to speed-dependent firing preferences is reminiscent of the similar subdivision observed in V2a interneurons and MNs[9,11,32–34]. In these neurons, there is a correlation between the recruitment order and differentiation order of the neurons: the ones that preferentially fire during fast swimming tend to be early-born neurons[9,34]. To examine whether such a developmental order was present between fast- and slow-type V1 neurons, electrophysiological recordings were made using animals in which early-born and late-born V1 neurons were distinguished by photoconvertible fluorescent protein Kaede (Fig. 1g). For the vast majority of cases (23 out of 25), the early-born neurons were found to be the fast-type (e.g., Fig. 1c). The remaining two cells were classified as hybrid-type (Supplementary Fig. 2). Preferred firings during fast swim in the early-born neurons are exemplified in Fig. 1h (left). In contrast, many of the late-born neurons tended to fire more reliably during slow swim (e.g., Fig. 1d). This tendency is exemplified in Fig. 1h (right). Figure 1i shows the summary of the classification, indicating that early-born V1 neurons prefer to become fast-type.

**Ablation of V1 neurons reduced cycle frequency in swimming.** We aimed to genetically ablate V1 neurons by expressing DTA in spinal V1 neurons. We generated Tg[*en1b*:loxP-RFP-loxP-DTA] and Tg[*hoxa4a/9a*:Cre] lines. In the latter, Cre was expressed in the spinal cord (plus a portion of the caudal hindbrain). In the compound transgenic fish (hereafter called En1-DTA), DTA was expressed in *en1b*-positive neurons in the spinal cord. Successful ablations of *en1b* neurons in the spinal cord were verified using the triple transgenic fish Tg[*en1b*:loxP-RFP-loxP-DTA], Tg[*hoxa4a/9a*:Cre], and Tg[*en1b*:GFP], in which green fluorescent protein (GFP)-positive *en1b* neurons in the spinal cord were almost completely absent (Fig. 2a and Supplementary Fig. 3a). The ablation of V1 neurons did not change the cell numbers of other types of neurons (Supplementary Fig. 3b–d).

We examined the fictive swimming of En1-DTA larvae and found that the frequency of VR bursts during swimming

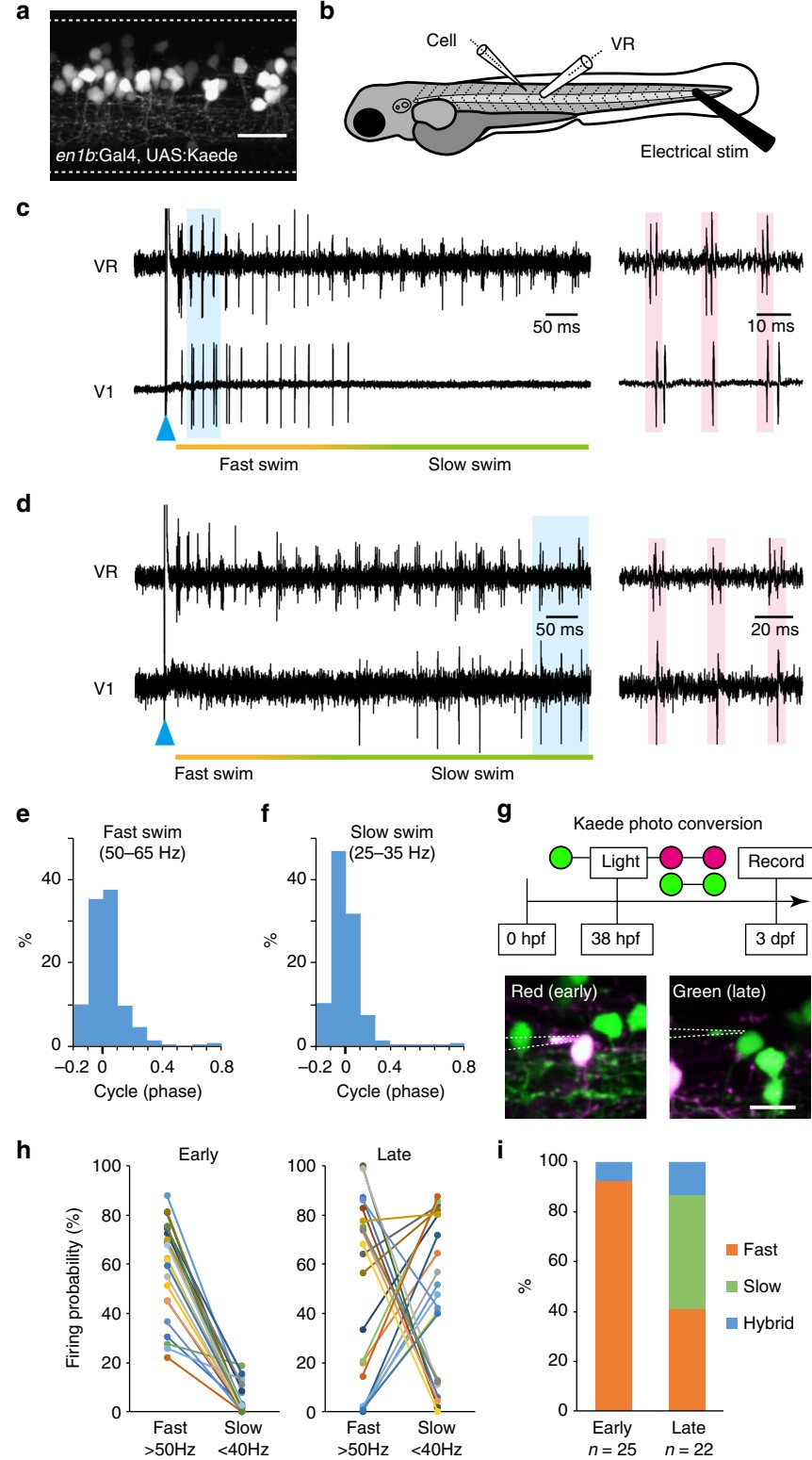

(swimming frequency) was markedly declined (Fig. 2b, c). This phenotype was most apparent at the fast-swim phase of swimming (swimming that immediately follows stimulation). For the quantification, we analyzed the swimming frequency of the swim bouts within 150 ms after the stimulation (hereafter called the initial phase of ES swim). During this period, control fish primarily performed fast swimming (>40 Hz; Supplementary Fig. 4a). As shown in Fig. 2d, swimming frequency during this period dramatically declined in En1-DTA fish. We also examined

swimming frequency during slow swim. For this purpose, we examined swimming frequency during swim bouts that occurred without ES. (These include spontaneously occurring swimming and swimming elicited by changing illumination intensity.) In these swim bouts (hereafter called Non-ES swim), swimming frequency was mostly within the slow-swim range (20–40 Hz; Supplementary Fig. 4b). As shown in Fig. 2e, En1-DTA fish showed a slight reduction in frequency during Non-ES swim. These results indicate that the absence of V1 neurons gave rise to

**Fig. 1** Firing patterns of V1 neurons in larval zebrafish. **a** A side view of the compound transgenic fish of Tg[*en1b*:Gal4] and Tg[UAS:Kaede]. The dashed lines indicate boundaries of the spinal cord. Scale bar, 20 μm. **b** A schematic illustration of the simultaneous recordings of a V1 neuron (loose-patch) and ventral root (VR). Fictive swimming was elicited by applying brief electrical stimulation near the tail. **c** An example of the recordings from the fast-type V1 neurons. The blue arrowhead shows the time point of electrical stimulation. In the right panel, the region shadowed in blue in the left panel is enlarged. **d** An example of the recordings from the slow-type V1 neurons. **e** Histogram of spike timings of fast-type V1 neurons during fast (50–65 Hz) swim (1387 swimming cycles from 29 cells). **f** Histogram of spike timings of slow-type V1 neurons during slow (25–35 Hz) swim (5782 swimming cycles from 10 cells). **g** Schematic diagram of the Kaede photo-conversion experiment (top). The bottom two panels show the recordings from early-born V1 neurons (red Kaede [shown in magenta], left) and late-born V1 neurons (green Kaede, right). Scale bar, 10 μm. **h** Firing probability of early-born V1 neurons (left, $n = 25$) and late-born V1 neurons (right, $n = 22$) in each cycle during fast (>50 Hz) and slow (<40 Hz) swim. Each colored circle represents each recorded cell. **i** Classification of the recorded V1 neurons. Out of the 25 early-born V1 neurons, $n = 23$ for fast-type and $n = 2$ for hybrid-type. Out of the 22 late-born V1 neurons, $n = 9$ for fast-type, $n = 10$ for slow-type, and $n = 3$ for hybrid-type. If the value of the firing probability during fast swim was more than double that during slow swim, the cell was considered a fast-type (and vice versa). If the difference was within the doubled value, the cell was considered a hybrid-type

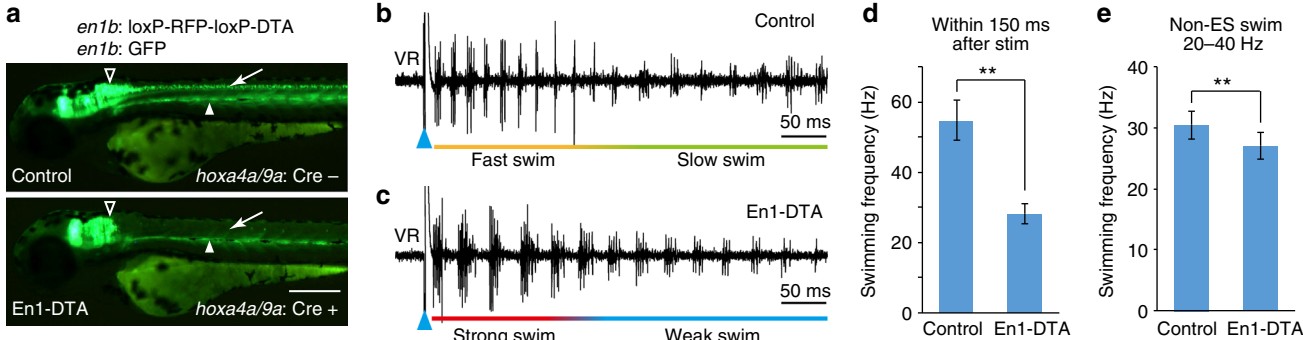

**Fig. 2** V1 ablation reduced cycle frequency in swimming. **a** Fluorescent images (green channel) of Tg[*en1b*:GFP] and Tg[*en1b*:loxP-RFP-loxP-DTA] fish with (bottom panel) or without (top panel) Tg[*hoxa4a/9a*:Cre]. Green fluorescent protein (GFP) expression in the spinal V1 neurons (arrow) is absent in the presence of Cre (En1-DTA), with GFP expression in the brain (triangle) and slow muscle cells in the middle region of the body (arrowhead) being intact. Scale bar, 250 μm. **b**, **c** Ventral root recordings of fictive swimming elicited by electrical stimulations (ESs) in a control (**b**) and an En1-DTA fish (**c**). **d** Swimming frequency of control and En1-DTA fish during the initial phase of ES swim (swimming elicited by electrical stimulation). Control: 54.8 ± 5.7 Hz, number of fish = 73. En1-DTA: 28.0 ± 2.9 Hz, $n = 57$. **$P < 0.01$ (two-tailed $t$ test, $P = 6.4 \times 10^{-62}$). **e** Swimming frequency of control and En1-DTA fish during Non-ES swim. Swim cycles with frequencies within 20–40 Hz were picked up and averaged. Control: 30.5 ± 2.3 Hz, $n = 73$. En1-DTA: 27.1 ± 2.2 Hz, $n = 57$. **$P < 0.01$ (two-tailed $t$ test, $P = 2.6 \times 10^{-14}$). Data are mean ± s.d.

general reduction in swimming speed, particularly during the fast phase of swimming.

In En1-DTA fish, swimming frequency was mostly stable during the entire period of swimming (around 27–28 Hz; Fig. 2d, e). However, the swimming during the initial phase was qualitatively different from that of the later phase, as inferred by the large-amplitude as well as dense VR bursts during the initial phase (Fig. 2c). This strongly suggests that the excitation level in the central nervous system was very high during this period, and like control fish, En1-DTA fish were also performing strong movements. As will be described in the following sections, MNs did indeed receive very strong excitation during this period. Strong movements of swimming during this period were also evident in motile fish. Upon sudden-touch stimulation, the swimming caused by large-amplitude muscular contractions was observed in both control and En1-DTA fish (Supplementary Movie 1). In En1-DTA fish, the duration of each bending, including the escape bend, was extremely prolonged.

In control fish, strong swim appears as a form of fast (high frequency) swim. In the case of En1-DTA fish, the frequency of the swimming (nearly constant at around 27–28 Hz) does not reflect the strength of the movement. Therefore, the term "strong swim" is used for En1-DTA fish when referring to their swimming during the initial phase after the stimulation (Fig. 2). The subsequent steady-state swimming is termed "weak swim."

**V1 ablation alters the recruitment patterns of slow-type MNs.** Next, we examined the phenotypes of En1-DTA fish at the level of individual neurons. We first focused on MNs.

MNs in larval zebrafish are anatomically classified into several groups[12]. Each of the classes of MNs shows distinctive speed-dependent firing patterns during swimming[10,12]. We wanted to examine phenotypes in an MN-class-dependent manner. For this purpose, fluorescent dye was introduced after the loose-patch recordings and the morphologies of the recorded MNs were then determined. In this study, MNs were grouped into two types: those that mainly innervate fast muscles (fast-type MNs) and those that mainly innervate slow muscles (slow-type MNs). With these criteria, primary MNs (PMNs) and the dvs-type of secondary MNs were classified as fast-type (Fig. 3a), while the iS-nc-type of secondary MNs was classified as slow-type (Fig. 3b)[12].

Figure 3c, d show representative examples of recordings from fast-type MNs. Fast-type MNs in control fish exhibited spiking activities mainly during fast swimming but became silent during slow swimming (Fig. 3c)[10,12]. This recruitment pattern was essentially unchanged in En1-DTA fish. Fast-type MNs mainly fired during strong swim and became silent during weak swim (Fig. 3d). A similar tendency was also observed in the comparison of the initial phase of ES swim (strong movements) and Non-ES swim (mostly weak swim; Supplementary Fig. 4). Fast-type MNs both in control and En1-DTA fish fired more reliably during the initial phase of ES swim (Supplementary Fig. 5a).

Figure 3e, f show representative examples of recordings from slow-type MNs. Slow-type MNs in control fish exhibited spiking activities mainly during slow swim (Fig. 3e)[10,12]. Strikingly, this recruitment pattern was completely altered in En1-DTA fish. Slow-type MNs consistently fired during the entire period of swimming including strong swim and weak swim (Fig. 3f). The

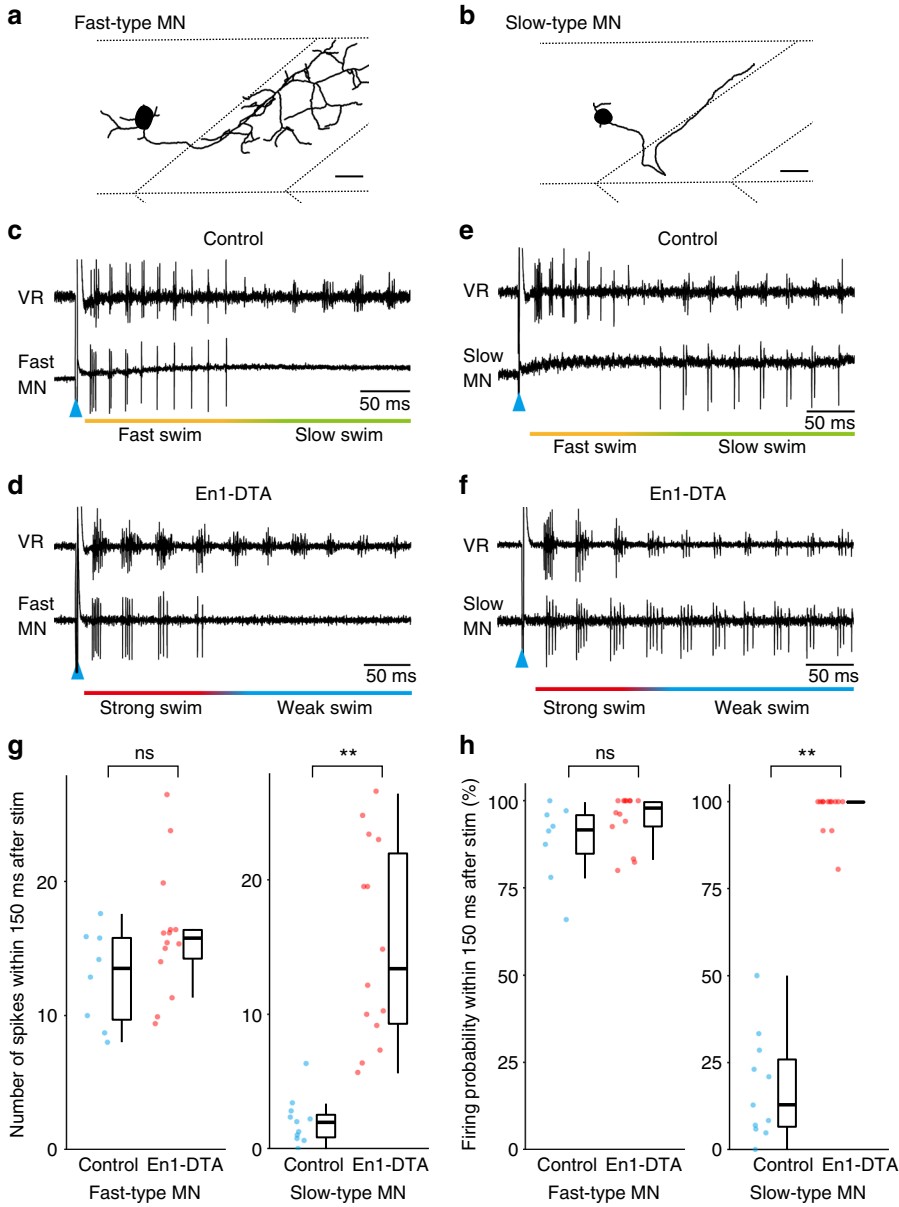

**Fig. 3** Activity of motoneurons (MNs) in control and En1-DTA fish during fictive swimming. **a** Typical morphology of fast-type MNs. The horizontal dotted line at the top indicates the dorsal boundary of the trunk muscles. The horizontal dotted line at the bottom indicates the boundary of the dorsally located muscles and ventrally located muscles. Scale bar, 20 μm. **b** Typical morphology of slow-type MNs. Scale bar, 20 μm. **c** Example of simultaneous recordings between fast-type MNs (loose-patch) and ventral root (VR) in control fish. **d** Example of simultaneous recordings between fast-type MNs and VR in En1-DTA fish. **e** Example of simultaneous recordings between slow-type MNs and VR in control fish. **f** Example of simultaneous recordings between slow-type MNs and VR in En1-DTA fish. **g, h** Numbers of spikes (**g**) and firing probability in each cycle (**h**) during fast/strong swim. Each circle represents each cell ($n = 8$ for fast-type MNs in control [$n = 6$ for PMNs, $n = 2$ for dvs]; $n = 14$ for fast-type MNs in En1-DTA [$n = 8$ for PMNs, $n = 6$ for dvs]; $n = 11$ for slow-type MNs in control; $n = 14$ for slow-type MNs in En1-DTA). **$P < 0.01$ (Mann–Whitney $U$ test, $P = 0.14$ [**g**, fast-type MNs], $P = 3.5 \times 10^{-5}$ [**g**, slow-type MNs], $P = 0.11$ [**h**, fast-type MNs] for fast-type MNs, $P = 1.2 \times 10^{-5}$ [**h**, slow-type MNs]). ns, not significant. Boxes represent the interquartile range (IQR) between first and third quartiles and the line inside represents the median. Whiskers denote the lowest and highest values within 1.5 × IQR from the first and third quartiles, respectively

alteration of recruitment patterns was also evident in a comparison between the initial phase of ES swim and Non-ES swim. In control fish, slow-type MNs preferentially fired during Non-ES swim (slow movements). In contrast, in En1-DTA fish, slow-type MNs equally fired during the initial phase of ES swim (strong movements) and Non-ES swim (weak movements) (Supplementary Fig. 5b).

Figure 3g, h show the quantitative analyses of spike numbers (Fig. 3g) and firing probability in each cycle (Fig. 3h) during the initial phase of ES swim. In the case of fast-type MNs, there

was no significant difference in either of the parameters between control and En1-DTA fish. (The fact that the total numbers of spikes were not significantly changed with the decrease of swimming frequency indicates that the fast-type MNs in En1-DTA fish exhibited greater numbers of spikes in each cycle Supplementary Fig. 5c). In contrast, in the case of slow-type MNs, there were huge differences in both of the parameters: En1-DTA fish exhibited large numbers of spiking activities with high probability in each cycle during strong swim.

We have thus far used Tg[*hoxa4a/9a*:Cre] as a Cre driver. In this line, Cre was expressed not only in the spinal cord but also in a portion of the caudal hindbrain. In order to show that the observed phenotype was primarily caused by the ablation of spinal V1 neurons, we used Tg[*hoxa9a*-3'enhancer:Cre] in which Cre expression was confined to the spinal cord (Supplementary Fig. 6a). We essentially obtained the same results using this driver (Supplementary Fig. 6b), indicating that the absence of spinal V1 neurons was responsible for the observed phenotype.

**V1 ablation reduces in-phase inhibition of slow-type MNs.** To gain insights into the cellular mechanisms underlying how V1 ablation led to the alteration of recruitment patterns in slow-type MNs, we performed voltage clamp recordings from slow-type MNs in order to measure input currents that the cells received. Slow-type MNs received rhythmic excitation during both fast and slow swimming (Fig. 4a)[10]. Slow-type MNs also received rhythmic inhibition during fast and slow swimming. The amplitude of inhibition during fast swimming was much larger than that during slow swimming (Fig. 4b)[10]. Examinations of the timing of inhibition indicated that the large-amplitude inhibition during fast swimming was mostly in-phase (Fig. 4b, middle)[10]. By contrast, in-phase inhibition was very small during slow swimming, and the amplitude of the inhibition was higher during the anti-phase period (Fig. 4b, right)[10].

In En1-DTA fish, the overall pattern of the excitation currents was not much different from that of control fish (Fig. 4c). In contrast, in the case of the inhibition, striking differences were observed between En1-DTA and control fish. Slow-type MNs received rhythmic inhibition whose amplitude was relatively stable during strong and weak swimming (Fig. 4d). This is in marked contrast to the pattern observed in Fig. 4b (control fish) in which the amplitude of inhibition during fast swim is much larger than that during slow swim. In addition, in-phase inhibition became almost negligible during strong swim in En1-DTA fish (Fig. 4d, middle). During weak swim, slow-type MNs in En1-DTA fish predominantly received anti-phase inhibition (Fig. 4d, right).

Figure 4e shows population data during the initial phase of ES swim that depicts the features described above. In control larvae, both the inhibitory and excitatory currents reached their maximum near time 0 (in-phase) in the swim cycle: the peak of the inhibition overlaps with the peak of the excitation. It should be noted that the peak inhibition timing coincided with the preferential firing timing of V1 neurons during fast swimming (Fig. 1e), suggesting that V1 neurons are the source of this in-phase inhibition in control larvae. In En1-DTA fish, in-phase inhibition was dramatically reduced, such that the peak of inhibition is located in the anti-phase. A dramatic decrease of in-phase inhibition is consistent with the idea that V1 neurons are indeed the source of in-phase inhibition.

The reduction and phase-shift of inhibition is summarized in Fig. 4f. In control larvae, the peak current during in-phase was larger than that during anti-phase. In contrast, in En1-DTA larvae, the peak was shifted to anti-phase. Figure 4f also shows that the amplitude of in-phase inhibition was greatly reduced in En1-DTA larvae.

We also performed quantitative analyses of the excitation currents during the initial phase of ES swim (Fig. 4g). In control larvae, the peak currents during the in-phase period were slightly larger than those during the anti-phase period. The amplitude of in-phase excitation became larger in En1-DTA larvae, and the difference of the amplitude between in-phase excitation and anti-phase excitation became more apparent (see Discussion).

We also performed voltage-clamp recordings from fast-type MNs. These neurons received strong phasic excitation and inhibition during fast/strong swimming both in control and En1-DTA larvae (Supplementary Fig. 7). In control larvae, the neurons received both in-phase and anti-phase inhibition (Supplementary Fig. 7b). In En1-DTA larvae, in-phase inhibition was negligible (Supplementary Fig. 7d).

The results described above, together with the firing patterns of V1 neurons (Fig. 1), strongly suggest that fast-type V1 neurons are responsible for providing in-phase inhibition both of slow-type and fast-type MNs during the initial phase of ES swim. This led us to examine whether direct synaptic connections were present between fast-type V1 neurons and slow/fast-type MNs by performing paired recordings. For slow-type MNs, we obtained 2 (out of 12) connected pairs. (An example is shown in Supplementary Fig. 8). For fast-type MNs, we obtained 2 (out of 7) connected pairs. (An example is shown in Supplementary Fig. 9).

**V1 ablation alters the recruitment patterns of V2a neurons.** Chx10-positive V2a neurons are the main sources of excitation that MNs receive during swimming[9,33,35]. V2a neurons also show speed-dependent firing preferences[9,11,33], leading us to speculate that V1 neurons may also be involved in setting up recruitment patterns in V2a neurons. We investigated the firing patterns of V2a neurons in control and En1-DTA fish.

For MNs, definite anatomical classification of MN types was possible. In the case of V2a neurons, however, there was no definite way to anatomically discriminate slow-type V2a neurons from fast-type V2a neurons. Instead, it is known that there are some relationships between neuronal locations and types of neurons: fast-type V2a neurons tend to be located in a dorsal region of the spinal cord, and slow-type V2a neurons tend to be located more ventrally[9,11,34,36]. Consequently, we performed loose-patch recordings from relatively ventrally located V2a neurons (Fig. 5a), expecting that slow-type V2a neurons would constitute a major part of our recorded samples. We indeed found neurons that preferentially fired during slow swim in control larvae, as can be seen in Fig. 5b and Supplementary Fig. 10a. In the case of En1-DTA larvae, many of the recorded V2a neurons fired throughout the episode, as the ones shown in Fig. 5c and Supplementary Fig. 10b, consistent with the idea that recruitment patterns of slow-type V2a neurons were altered. To perform statistical analyses, we took recordings from a large number of ventrally located V2a neurons ($n = 84$, control; $n = 60$, En1-DTA), and compared the results as populations. Figure 5d, e show the numbers of spikes and the firing probability in each cycle during the initial phase of ES swim (fast/strong swim). For both of the parameters, ventrally located V2a neurons in En1-DTA larvae exhibited elevated activities. The results indicate that ventrally located V2a neurons in En1-DTA larvae did indeed become more active during strong swim.

We also performed classification of the recorded V2a neurons by examining their firing preferences during the initial phase of ES swim (fast/strong swim) and Non-ES swim (mostly slow/weak swim). As depicted in Supplementary Fig. 11, the percentage of slow/weak-type V2a neurons was greatly reduced in En1-DTA fish.

**V1 ablation alters synaptic inputs onto slow-type muscles.** We have thus far reported that slow-type V2a neurons and MNs became more active during strong swim in En1-DTA fish. As a consequence, one would expect that slow muscles would receive increased synaptic inputs during strong swim. We tested this idea

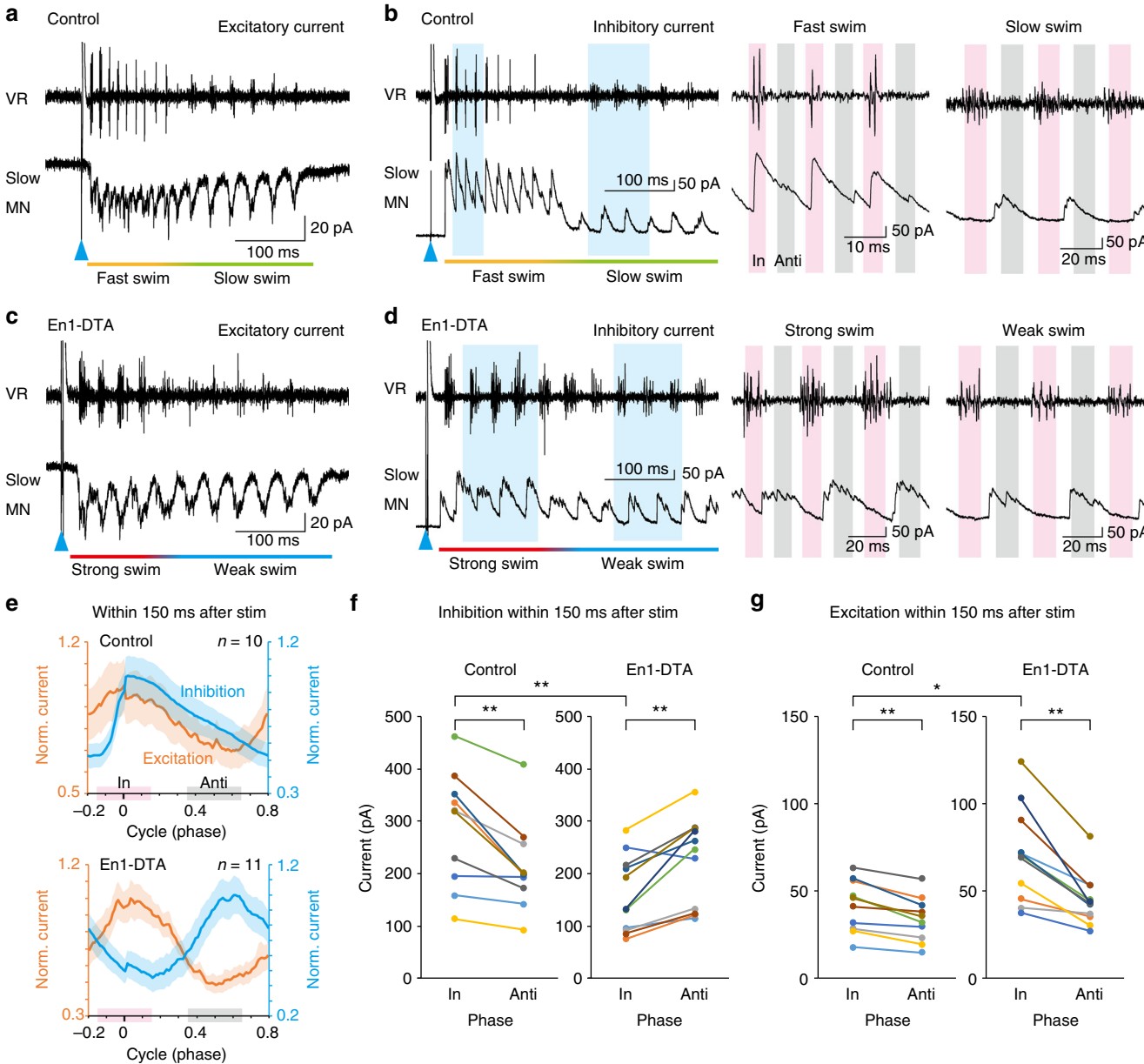

**Fig. 4** Voltage-clamp recordings from slow-type motoneurons (MNs) in control and En1-DTA fish. **a** Example of simultaneous recordings between slow-type MNs (voltage-clamp) and ventral root (VR) in control fish. The cell was held at −75 mV to reveal excitatory currents. **b** Example of simultaneous recordings between slow-type MNs and VR in control fish. The cell was held at +10 mV to reveal inhibitory currents. The right two panels show the enlargements of the regions shaded in blue. "In" represents the in-phase period (from −0.15 to +0.15 in the cycle, pink). "Anti" represents the anti-phase period (from 0.35 to 0.65, gray). **c** Example of simultaneous recordings between slow-type MNs and VR in En1-DTA fish. The cell was held at −75 mV to reveal excitatory currents. **d** Example of simultaneous recordings between slow-type MNs and VR in En1-DTA fish. The cell was held at +10 mV to reveal inhibitory currents. **e** Phase analysis of excitatory and inhibitory currents during fast/strong swim. Normalized currents (±s.e.m.) are shown. Top panel, control fish; bottom panel, En1-DTA. **f**, **g** Peak inhibitory (**f**) and excitatory (**g**) currents during fast/strong swim in control (left, $n = 10$) and En1-DTA (right, $n = 11$) fish. "In" represents the in-phase period. "Anti" represents the anti-phase period. Each circle represents each cell. **$P < 0.01$; *$P < 0.05$ (comparison between control and En1-DTA, Mann–Whitney $U$ test, $P = 0.0080$ [**f**], $P = 0.013$ [**g**]; comparison between in-phase and anti-phase, Wilcoxon signed-rank test, $P = 0.0020$ [**f**, control], $P = 0.0029$ [**f**, En1-DTA], $P = 0.0020$ [**g**, control], $P = 0.00098$ [**g**, En1-DTA])

by performing simultaneous voltage-clamp recordings from slow muscles and fast muscles during fictive swimming (Fig. 6a, b)[37].

We primarily analyzed Non-ES swim. In these swim bouts, swimming frequency was, on average, low (20–40 Hz). However, the frequency was not uniform and was occasionally high (>40 Hz; Supplementary Fig. 4b). Figure 6c shows an example of recordings in control larvae. During the two swim bouts, swimming speed was higher in the initial part (fast). In these phases of swimming, the synaptic current recorded in the slow

muscle was negligible. As the swimming speed decreased, synaptic currents recorded in the fast muscle became diminished, and the slow muscle predominantly received synaptic currents (slow)[37] (see also Supplementary Fig. 12 for the corresponding recording in a slow-type MN during Non-ES swim). Figure 6d shows an example of recordings in En1-DTA larvae. During one swim bout, the larva appeared to perform three instances of strong swimming (strong), as inferred by the large-amplitude synaptic currents in the fast muscles. Strikingly, during these

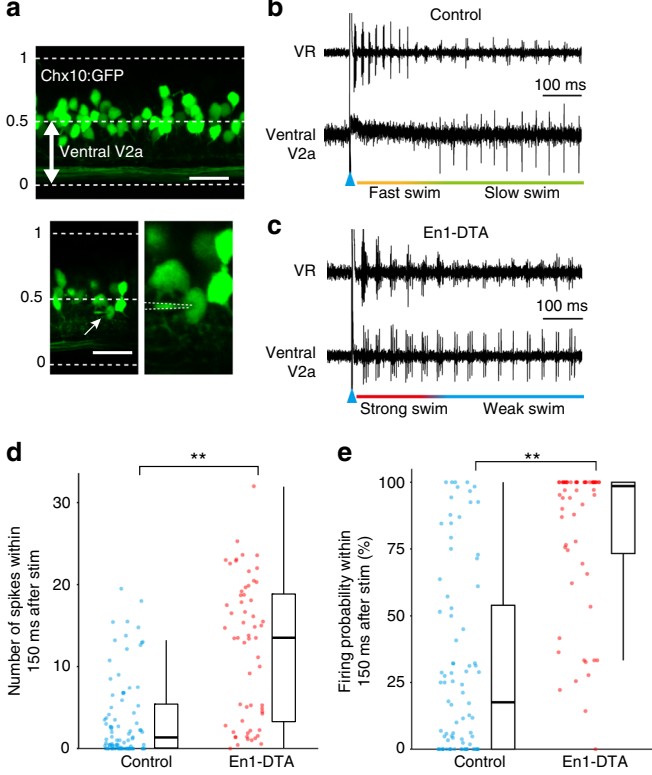

**Fig. 5** Activity of ventrally located V2a neurons in control and En1-DTA fish. **a** Lateral view of the spinal cord of Tg[*chx10*:GFP]. The horizontal dashed lines indicate the dorsal and ventral boundaries of the spinal cord. Ventrally located V2a neurons (the position from 0 to 0.5) are the subjects of the recordings (top panel). The bottom two panels show the images of a loose-patch recording. Scale bar, 20 μm. **b** An example of simultaneous recordings between ventrally located V2a neurons (loose-patch) and ventral root (VR) in control fish. **c** An example of simultaneous recordings between ventrally located V2a neurons (loose-patch) and VR in En1-DTA fish. **d**, **e** Numbers of spikes (**d**) and firing probability in each cycle (**e**) during fast/strong swim. Each circle represents each cell ($n = 84$ for ventrally located V2a neurons in control; $n = 60$ for ventrally located V2a neurons in En1-DTA). **P < 0.01 (Mann–Whitney $U$ test, $P = 1.3 \times 10^{-10}$ [**d**], $P = 3.7 \times 10^{-14}$ [**e**]). Boxes represent the interquartile range (IQR) between first and third quartiles and the line inside represents the median. Whiskers denote the lowest and highest values within 1.5 × IQR from the first and third quartiles, respectively

strong swimming periods, the slow muscle also received large-amplitude synaptic currents. For the remaining periods, the slow muscle predominantly received synaptic currents (weak). Overall, the switching pattern observed in control larvae was absent. Figure 6e shows the results of the quantitative analysis. During the swim cycles in which the fast muscles received large-amplitude currents, the slow muscles in En1-DTA fish received larger-amplitude currents. These results indicate that the normal pattern of synaptic inputs onto fast and slow muscles was altered in En1-DTA fish.

## Discussion

We have revealed two functions of V1 neurons during swimming: the regulation of cycle frequency and the suppressing activities of slow components during fast/strong movements.

The first is the long-proposed role of V1 neurons in aquatic vertebrates[27,28]. It has been speculated that V1 neurons help terminate the firing activities of CPG neurons and MNs in each

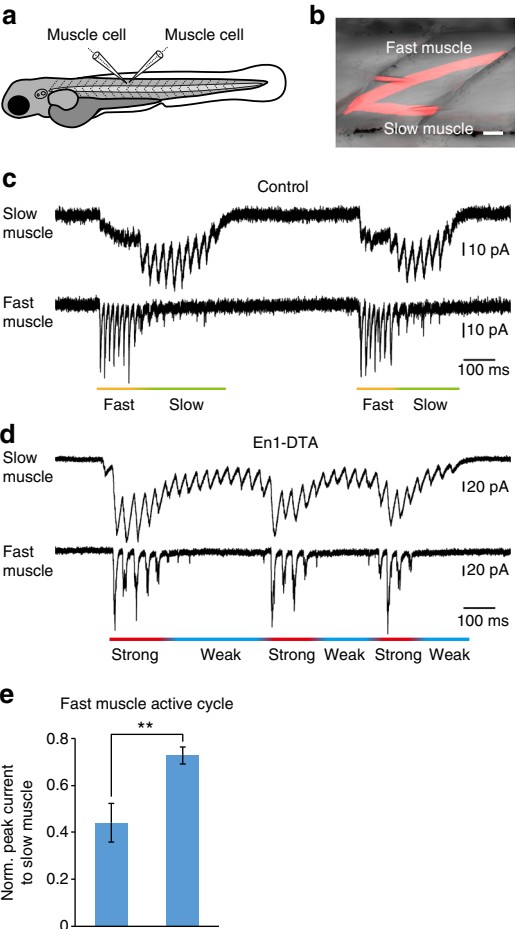

**Fig. 6** Voltage-clamp recordings of fast and slow muscles in control and En1-DTA fish. **a** A schematic illustration of the simultaneous voltage-clamp recordings of two muscle cells (one for a fast muscle, the other for a slow muscle). Neuro-muscular transmission was partially blocked in these experiments, and portions of synaptic currents remained in the recording conditions. **b** An image after a recording. Slow and fast muscles were morphologically distinct and could be easily distinguished. Scale bar, 20 μm. **c** Example of simultaneous voltage-clamp recordings in control fish. **d** Example of simultaneous voltage-clamp recordings in En1-DTA fish. **e** Normalized currents in the slow muscle cells in the cycle when the fast muscle received strong inputs (>50% of the maximum). Five paired recordings were performed. Control: 0.44 ± 0.08, En1-DTA: 0.73 ± 0.04. Data are mean ± s.d. **P < 0.01 (two-tailed $t$ test, $P = 0.00091$)

swim cycle by providing in-phase inhibition to these neurons[28,38]. In this scheme, inactivation of V1 neurons is expected to result in a prolongation of the cycle period in swimming. Here we provided genetic evidence. Our voltage-clamp recordings from MNs in En1-DTA fish showed that in-phase inhibitions of MNs dropped to a negligible level in both fast-type and slow-type MNs (Fig. 4d and Supplementary Fig. 7d). Furthermore, we found direct synaptic connections between fast-type V1 neurons and fast/slow-type MNs (Supplementary Figs. 8 and 9). These results indicate that V1 neurons are indeed the source of in-phase inhibition. As a consequence of this inhibition, cycle periods became prolonged (Fig. 2), with MNs exhibiting an elevated number of spikes in each cycle (Supplementary Fig. 5c). This is consistent with the results obtained in mice[30].

The regulation of swimming frequency by V1 neurons is at work both during fast and slow swim (Fig. 2d, e). During fast

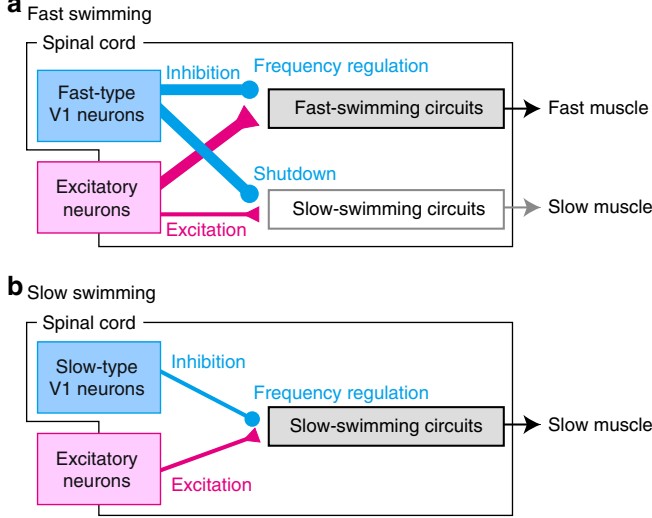

**Fig. 7** Proposed model of the functions of V1 neurons. **a** During fast swimming, excitatory neurons (pink) provide strong phasic excitation to fast-swimming circuits (thick pink line). Excitatory neurons (pink) also provide moderate phasic excitation to fast-swimming circuits (thin pink line). Fast-type V1 neurons provide strong phasic inhibition to both of the circuits (thick blue lines). The strong phasic inhibition of fast-swimming circuits plays a role in frequency regulation. The strong phasic inhibition of slow-swimming circuits plays a role in shutting down the activities of the component neurons. **b** During slow swimming, excitatory neurons (pink) provide phasic excitation to slow-swimming circuits (pink line). Slow-type V1 neurons provide phasic inhibition to the circuit (blue line). The phasic inhibition plays a role in frequency regulation

swim, fast-type V1 neurons perform this function by acting on fast-swimming circuits (Fig. 7a), while during slow swim, slow-type V1 neurons perform this function by acting on slow-swimming circuits (Fig. 7b). The present study showed that the phenotype is more severe during the fast-swim period (Fig. 2), suggesting that frequency regulation by V1 neurons is particularly important during fast swim. Defects in frequency regulation during the fast-swim period resulted in low-frequency tail beat with very large body bend (Supplementary Movie 1; decoupling of tail-beat frequency and the amplitude of body bend are also seen during struggling[39]).

The second function we revealed is completely new. V1 neurons are crucial for suppressing activities of slow components during fast/strong movements. It is known that slow-type V2a neurons and slow-type MNs are often inactive during fast swimming in larval zebrafish[9–12]. To account for this phenomenon, two neuronal mechanisms are possible. One is that these slow-component neurons receive few excitatory inputs during fast swimming. The other is that they receive excitation together with very strong inhibition that surpasses excitation. Voltage-clamp recordings in both a previous study[10] and the current study have revealed that slow-type MNs do receive rhythmic excitatory inputs during fast swimming, suggesting that the latter scenario is likely. Indeed, voltage-clamp recordings for measuring inhibitory currents have shown that slow-type MNs receive very strong in-phase inhibition during fast swimming[10] (Fig. 4b). The peak timing of the inhibition coincides with the timing of the spiking activity of fast-type V1 neurons (Fig. 1e), strongly suggesting that fast-type V1 neurons are the source of the in-phase inhibition. Our genetic ablation studies provided a clear answer for the suppressing role of V1 neurons. In En1-DTA fish, strong in-phase inhibition was almost completely absent (Fig. 4d, e), and slow-type MNs were vigorously active during strong swimming

(Fig. 3f). These data are consistent with the idea that strong in-phase inhibition coming directly from V1 neurons surpasses the excitatory inputs, preventing slow-type MNs from spiking in control fish. The discussion described above assumes that there are direct synaptic connections between fast-type V1 neurons and slow-type MNs, which indeed exist (Supplementary Fig. 8).

As for slow-type V2a neurons, voltage-clamp recording data are not available. However, paired recordings showed the presence of direct connections between aINs (V1 neurons) and dINs (likely V2a neurons) in frog tadpoles[28], suggesting that the silencing mechanisms of slow-type V2a neurons by V1 neurons are the same as the one discussed above for slow-type MNs. In addition to direct inhibition from V1 neurons, inhibitory effects may retrogradely come from MNs through gap junctions[40].

Recruitment of slow-type V2a neurons during strong swimming appears to affect excitatory inputs that slow-type MNs receive. In En1-DTA fish, in-phase excitatory inputs during strong swimming were significantly increased compared to control fish (Fig. 4g). Elevated firing activities of slow-type V2a neurons during strong swimming were likely to contribute to the increase of the excitatory inputs in slow-type MNs.

Figure 7a depicts the dual functions of fast-type V1 neurons during fast swimming. As noted above, fast-type V1 neurons act on fast-swimming circuits and regulate swimming frequency. Concurrently, fast-type V1 neurons act on slow-swimming circuits and play a role in shutting down their activities. The dual functions of fast-type V1 neurons can be explained by the difference in the strength of excitatory inputs that fast- and slow-component neurons receive. In fast-type MNs, these neurons receive very strong (>100 pA) in-phase excitation during fast swim[10] (Supplementary Fig. 7a). This is large enough for fast-type MNs to fire, even in the presence of very strong (hundreds of pA) in-phase inhibition coming from fast-type V1 neurons[10] (Supplementary Fig. 7b). The strong in-phase inhibition instead restricts the firings of fast-type MNs in a very narrow time window, thereby making a rapid cycle period during fast swim possible. In slow-type MNs, they receive moderate (tens of pA) in-phase excitation during fast swim (Fig. 4a). This itself would be large enough for slow-type MNs to fire. However, because of very strong in-phase inhibition (hundreds of pA, Fig. 4b)[10] from fast-type V1 neurons, the spiking-activity of slow-type MNs is almost completely suppressed. The same action by fast-type V1 neurons is likely to be working on slow-type V2a neurons. The present study mostly focuses on the functions of fast-type V1 neurons during fast swimming. Detailed functional analyses of slow-type V1 neurons during slow swimming will be reported in future studies.

What is the physiological significance of the silencing described above? It is probably related to mechanical demand for muscle contraction and relaxation. As a consequence of the silencing of slow-type MNs, slow muscles are deactivated in fish during very fast swimming[10,12,13,15,37]. Presumably, the participation of slow muscles is counterproductive to very fast swimming, as slow muscles are tuned to slower movements with slow contraction and relaxation times[41].

In the original "size principle," deactivation or derecruitment of slow-type MNs (hence, slow-type muscle units) is not considered. However, several lines of evidence imply that orderly recruitment of motor units does not always occur in mammals[16], suggesting the presence of neuronal mechanisms for selective recruitments. For example, selective recruitment of predominantly fast muscle fibers (gastrocnemius) with the silence of slow muscle fibers (soleus) was reported during very rapid paw shakes in cats[42]. Within the same muscles of mixed fiber types, shifts in muscle fiber recruitment are suggested to occur with derecruitment of slow muscle fibers before the faster fibers during

very high-frequency movements in humans[17]. In these cases, deactivation of slow muscle fibers is likely to be related to the mechanical demand for fast contraction and relaxation during these fast movements. The neuronal basis of the deactivation of slow motor units in mammals is unknown, but it is possible that inhibitory inputs provided by V1 neurons play a similar role. Mammalian V1 neurons consist of several subclasses including Renshaw cells[26]. Interestingly, it has been shown that the recurrent inhibitory influence of Renshaw cells differs between motor unit types, with fast units being less inhibited than slow units, raising the possibility that Renshaw cells are involved in suppressing slow-type MNs[43]. We expect that future studies in mammals will uncover whether V1 neurons, including Renshaw cells, play an active role in the suppressing activities of slow-component neurons during fast movements.

## Methods

**Animals.** Zebrafish adults, embryos, and larvae were maintained at 28.5 °C. Experiments were performed at room temperature (23–28 °C). All procedures were performed in compliance with the guidelines approved by the Animal Care and Use Committees of the National Institutes of Natural Sciences. Animals were staged according to hours post fertilization (hpf) or dpf.

The following transgenic strains were used in the present study: Tg[en1b:Gal4][31], Tg[UAS:Kaede][44], Tg[mnr2b:GFP][45], Tg[chx10:GFP][9], Tg[hoxa4a/9a:Cre], Tg[hoxa9a-3'enhancer:Cre], Tg[en1b:loxP-RFP-loxP-DTA], and Tg[en1b:GFP] (this study). For the generation of Tg[hoxa4a/9a:Cre], an enhancer sequence of the hoxa4a gene (approximately 5 kb in length; 5' sequence, TTTTTGTTTACTTTT TAGTG; 3' sequence, CATAAAAATTAGAATTGTTG) and an enhancer sequence of the hoxa9a gene (approximately 4 kb in length; 5' sequence, AGCTAGCTACA AGCAGCAGA; 3' sequence, GTTAATTGTTTCCACTGGAT) were isolated from zK25E11 BAC. For the generation of Tg[hoxa9a-3'enhancer:Cre], an enhancer sequence located downstream of the hoxa9a gene (approximately 4 kb in length; 5' sequence, CTCGAGCTGGAGAAAGAGTT; 3' sequence, ACGAAGAAGTATA TGAATTC) was isolated from zK25E11 BAC. The hoxa4a enhancer, the hsp70 promoter, Cre-mCherry-NLS[46], BGH poly(A), and the hoxa9a enhancer were subcloned into a Tol2-based plasmid[47], and transgenic fish were generated. Tg[en1b:loxP-RFP-loxP-DTA] and Tg[en1b:GFP] transgenic fish were generated using the CRISPR/Cas9-mediated knock-in method with the hsp70 promoter[31].

**Photoconversion of Kaede.** Photoconversion of Kaede was performed by broadly illuminating embryos with violet light (425DF60 nm) using a fluorescence-dissecting microscope (FLIII, Leica). Embryos at 38 hpf in their chorions were illuminated for a few minutes until the fluorescence from Kaede became completely red. During illumination, the position of the embryos was occasionally changed with forceps.

**Immunohistochemistry.** Whole-mount immunohistochemistry using the rabbit anti-Evx2 antibody[46] was performed with a standard procedure.

**Electrophysiology.** Loose-patch, whole-cell, and VR recordings[9,44,48] were performed as follows. Recordings were carried out using 3-dpf larvae. Larvae were immobilized by soaking them in the neuromuscular blocker d-tubocurarine (0.1 mg per ml in distilled water) for 5–15 min, and they were then pinned through the notochord to a Sylgard-coated, glass-bottomed dish with short pieces of fine tungsten pins. Animals were then covered with extracellular recording solution that contained (in mM) 134 NaCl, 2.9 KCl, 1.2 MgCl$_2$, 2.1 CaCl$_2$, 10 HEPES, 0.015 d-tubocurarine, and 10 glucose, adjusted to pH 7.8 with NaOH. The skin covering the midbody was removed with a pair of forceps. Then muscle fibers of one segment were carefully removed manually with a tungsten needle. For all electrophysiology experiments, the preparations were observed using a water immersion objective (×40; NA, 0.80; Olympus) on an upright microscope (BX51WI; Olympus) fitted with differential interference contrast optics. Neurons located in the midbody segments (segments 10–15) were targeted for recordings. We used the following transgenic fish for the targeted recordings: Tg[en1b:Gal4] and Tg[UAS:Kaede] for V1 neurons, Tg[mnr2b:GFP] for MNs, and Tg[chx10:GFP] for V2a neurons. VR recordings were made just 1–2 segments caudal to the recording site of neurons. Electrodes for VR recordings (tip diameter, 30–60 μm) and loose-patch recordings (resistance, ~14 MΩ) were filled with the extracellular recording solution. Patch electrodes (resistance, ~14 MΩ) were filled with intracellular solution. The intracellular solution for current clamp recording contained (in mM) 119 K-gluconate, 6 KCl, 2 MgCl$_2$, 10 HEPES, 10 EGTA, and 4 Na$_2$ ATP at 290 mOsm and adjusted to pH 7.2 with KOH. The calculated liquid junction potential was 15 mV. The calculated chloride-reversal potential was −52 to −53 mV. The intracellular solution for voltage clamp recording contained (in mM) 140 CsMeSO$_4$, 1 QX314-Cl, 1 TEA-Cl, 3 MgCl$_2$, 10 HEPES, 1 EGTA, 4 Na$_2$-ATP, adjusted to pH 7.2 with CsOH[10]. For measuring excitatory currents, cells were held at −75 mV. For

measuring inhibitory currents, cells were held at +10 mV. These values represent the calculated chloride ion and cation reversal potentials, respectively. Values were corrected for a calculated junction potential of −11 mV (pClamp 10; Molecular Devices). Electrophysiological recordings were performed using MultiClamp700B amplifiers and digitized with Digidata1440A (Molecular Devices). Neurons were labeled with 0.01% Alexa Fluor 597 or 647 hydrazide (Thermo Fisher Scientific) in patch solution. Fictive locomotion was elicited either by applying a brief electric shock (stimulus strength of 7–20 V for a duration of 0.2–1.0 ms) or by changing the illumination intensity. Fictive swimming also occurred spontaneously. For voltage-clamp recordings from muscle cells, immobilization of larvae was performed with relatively low-concentration d-tubocurarine (3 μM), such that some levels of neuro-muscular transmission were present[37]. The fast and slow muscle cells were held at −90 and −67 mV, respectively. After the whole-cell recordings of MNs or muscles, fluorescent images were acquired with a FV300 confocal unit with a 543- or 633-nm laser (Olympus).

For paired recordings between V1 neurons and post-synaptic target neurons, Tg[en1b:Gal4] and Tg[UAS:Kaede] transgenic fish were used. Muscle fibers of 2–3 segments were removed. Fluorescent V1 neurons were targeted for extracellular recordings. Possible post-synaptic MNs located in the ventral spinal cord near the recorded V1 neurons (less than one segment away) were randomly targeted for intracellular (current clamp) recordings. The nanostimulation method[49] was used to elicit action potentials in extracellularly recorded V1 neurons. Briefly, square currents (1–10 nA in amplitude; 50–200 ms in duration) were injected into the recording pipette during extracellular recordings. To verify that the recorded neurons were MNs, we examined the morphology of the recorded cells; in the case of MNs, toned axons at the lateral edge of the ventral spinal cord were visible. Type of MNs (fast or slow) was determined by examining the firing patterns of the cell during fictive swimming elicited by ES.

**Data analysis.** Electrophysiological data were analyzed with DataView (software by William Heitler, University of St. Andrews) and Excel (Microsoft). VR recordings were rectified and smoothened. To detect each instance of VR activity, a threshold value was set by a visual inspection. For the phase analysis, the middle time point of a VR activity was assigned a phase value of 0 and that of the next VR activity was assigned a phase value of 1. For determining the frequency of swimming, duration between time point 0 and 1 was defined as a cycle period. Swimming frequency was the inverse of the cycle period. For the presentation of swimming frequency shown in Fig. 2d, e, the average value in each recorded fish was first calculated, and the overall average was then calculated. For Fig. 2d, e, at least three swimming episodes were analyzed for each fish.

For the presentation of spike numbers and firing probability (Figs. 3g, h and 5d, e; Supplementary Figs. 5 and 11a, b), more than three swimming episodes were examined for each cell for the vast majority of cases. In some cases (representing <10% of the samples), only one or two swimming episodes were analyzed due to the early termination of the recordings.

For the quantitative analysis of input currents received by MNs (Fig. 4e–g), we examined at least three swimming episodes for each cell. To avoid the potential confounding influence of direct sensory inputs to MNs following the electrical stimulus, we did not analyze the first motor burst. Peak excitatory current and peak inhibitory current were the averaged excitatory or inhibitory maximum of every event analyzed (Fig. 4f, g). The phase timing between −0.15 and 0.15 is regarded as in-phase, and the timing between 0.35–0.65 is regarded as anti-phase. For phase analysis of excitation and inhibition received by MNs (Fig. 4e), we analyzed synaptic currents from phase 0 to 1, divided into 100 equal segments. The averaged excitatory and inhibitory currents across cells were normalized to the peak value of the currents.

To analyze the synaptic currents in muscle cells (Fig. 6e), the peak currents in each swimming cycle were normalized to the max. value of the current recorded from each muscle cell. In each fish, we averaged normalized peak currents in slow muscle cells during the swimming cycles in which the fast muscle cells received large-amplitude currents (>0.5 of the max. current). The value thus obtained was averaged for five fish, and the result is presented. For each fish, at least four swim episodes occurred (>70 swim cycles were present). Among them, there were at least six swim cycles in which fast muscles received large currents.

**Statistics.** Statistical analyses were performed using Excel (Microsoft) or R (version 3.5.0) (R Core Team, 2018). Statistical comparisons between independent groups were performed using either the Mann–Whitney U test or Student's t test. Comparisons between related groups were performed using Wilcoxon signed-rank tests. Significance was set at P < 0.05 (*P < 0.05, **P < 0.01). Results are presented as the mean ± s.d. (standard deviation), except for Fig. 4e where the result is presented as the mean ± s.e.m. (standard error of the means).

**Reporting summary.** Further information on research design is available in the Nature Research Reporting Summary linked to this article.

## Data availability
The data that support the findings of this study are available from the corresponding author upon reasonable request.

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

## Acknowledgements

We are grateful to Dr. Asakawa and Dr. Kawakami for providing the Tg[mnr2b:GFP] strain. We also thank Dr. Ono, Dr. Seki, Dr. Nishimaru, Dr. Matuda, Dr. Miyasaka, Dr. Yoshihara, and Dr. McLean for their helpful discussions. This work was supported in part by grants from the Ministry of Education, Culture, Sports, Science and Technology of Japan.

## Author contributions

Both the authors conceived and designed the study and wrote the manuscript. YK performed the major parts of the experiments. SH generated the transgenic fish lines.

## Additional information

**Competing interests:** The authors declare no competing interests.

