## [Peer Review File · Nature Communications]

Reviewers' Comments:

Reviewer #1:

Remarks to the Author:

This is a sound and well executed study exploring the role of a class of spinal interneuron (V1 INs) in mediating the interaction between fast and slow component neurons in zebrafish larvae. The analysis relies mainly on genetic ablation of V1 interneurons combined with electrophysiological recordings. The results show that some V1 INs are active during fast locomotion and others are active during slow swimming in addition to some that are active during both slow and fast locomotion. In control animals, neurons (MNs and V2a INs) that are active at slow swim frequencies are silenced during fast/stong locomotion. This suppression is not seen after ablation of V1 INs. The authors conclude that V1 INs play a crucial role in the mediating the primacy of fast over slow locomotor activity in zebrafish larvae.

The manuscript is clear and the results are well illustrated – it provides novel circuit mechanisms underlying the interaction between neurons active during fast and slow locomotor movements. This is a strong study suitable for publication in Nature Communications after revisions. The following concerns, comments and questions are meant to help providing more clarity of the results and their interpretation in a broader context.

1) The interpretation of the data is made very difficult because of the change in the frequency after ablation of V1 INs. The following issues need to be addressed:

- i) Why do fast neurons become recruited at slow frequencies in V1 ablated fish?
- ii) Is the lack of suppression of slow neurons activity simply due to the fact that the swimming does not reach frequencies as high as in control?
- iii) Is the change seen only due to absence of V1 INs or to an unmasking of the effect of other neurons?
- iv) Does ablation of V1 affect escape behavior?

2) Is the suppression simply due to the use of electrical stimulation that enables recruitment of fast (maybe escape related) neurons or is it the result of an increase in swimming frequency? This could be addressed by examining spontaneous swimming with increased frequency to see if during a spontaneous increase in frequency there is also a suppression of the activity of slow swimming neurons.

3) Related to the previous point, the circuit mechanism proposed is reminiscent of what has been shown in adult zebrafish. In a study by Song et al (Current Biology, 2015), it was shown that escape momentarily suppresses swimming via activation of an inhibitory class of interneuron possibly V1. This study should be cited and discussed in the context of primacy of escape over swimming.

4) How does the ablation of V1 neurons in the hindbrain contribute to the observed effects? The authors should at least acknowledge this in the discussion.

5) The paper would gain in strength if the authors could provide some connectivity data supporting the schematic in Figure 7. Do fast V1 IN connect to all slow neurons or only to some? Do slow V1 INs connect or not to fast neurons? Such results could strengthen the authors' conclusions. Without such data the proposed role of V1 remains speculative.

6) To assess the pattern of activity of the different neurons during fast vs slow swimming, it would be better to show current clamp data showing the change in membrane potential of fast vs slow V1, MNs and V2a's during swimming in control and in ablated fish (MNs and V2a's). The data as shown only inform about the firing of neurons. These neurons could still be receiving subthreshold rhythmic

excitation; therefore, it is necessary to provide current clamp data.

7) Figure 2: It is confusing to use both fast vs slow swim and strong vs weak swim. The main effect of V1 ablation is that there is a decrease in the frequency of stimulation-induced locomotion with little effect on the frequency of spontaneous swimming. The lack of suppression of the activity of slow/weak neurons could be just the reflection of fact that escape or fast swimming is compromised.

8) Figure 3: Please provide a comparison of the minimum recruitment frequency of the fast vs slow MNs in control and in V1 ablated fish. Again, why do fast MNs become recruited at slow swimming frequency? According to the available data in larval zebrafish, fast neurons should not be recruited at frequencies below 30 Hz. Please clarify.

Please show the distribution (box and whisker) for the right graph in Figure 3H (En1-DTA, slow MN)

9) Figure 9: The fact that slow MNs are receiving rhythmic on-cycle excitatory current and tonic inhibition. This shows that the excitatory interneurons driving locomotion are still active and provide rhythmic excitatory drive to slow MNs. These results also indicate that the effect of suppression of slow swimming is mainly occurring at the level of MNs and not at the level of INs (V2a's). This is similar to what was shown in adult zebrafish by Song et al (Current Biology, 2015).

The inhibition of firing of V2a INs shown in Figure 5 could be the results of electrical coupling that allows for the propagation of inhibition from MNs retrogradely to V2a INs (see Song et al., Nature, 2016). This should be at least mentioned in the discussion.

10) Figure 6: Why do slow muscle fibers receive a strong excitatory current during fast swimming? Shouldn't they be quiescent? If the activity of slow MNs is suppressed during fast swim, there shouldn't be any excitatory current in slow muscle fibers. The recording shows that slow muscle fibers receive excitation during fast locomotion perhaps because the slow muscles are also innervated by fast MNs. I think it is more appropriate to compare the current (Figure 6e) only in the portion of the recordings where the fast muscle fibers are quiescent.

It seems that slow muscle fibers are also innervated by fast MNs and as such their activity is not suppressed during strong/fast locomotion. The existence of inward current in slow muscle fibers during activity of fast MNs in the absence of slow MN activity should be mentioned and discussed.

11) In both the introduction and discussion, the authors should mention/discuss work from juvenile/adult zebrafish on the recruitment of MNs and V2a INs showing that there is no suppression of the activity of slow neurons during fast swimming and that this suppression only occurs during escape.

Reviewer #2:

Remarks to the Author:

Kimura and Higashijima present here an amazing piece of work - this is really a historical study. Congratulations!

Since the 80s, the motor field represented by N. Dale, A. Roberts, J. Buchanan and S. Grillner had identified in *Xenopus* and lamprey that in-phase inhibition onto motor neurons could be critical to set the locomotor frequency.

The work of Profs. Alan Roberts, Weng Chang Li and Steve Soffe in *Xenopus* identified glycinergic inhibitory neurons that provide in-phase inhibition onto motor neurons to set locomotor frequency. However, from the work in the *Xenopus* and lamprey, the genetic identification of these interneurons was unknown.

Previous work from Dave McLean and collaborators had shown that spinal V1 interneurons are recruited in a speed-dependent manner. The authors confirm this finding in the first figure.

In this study combining an elegant genetic ablation of V1 interneurons in the spinal cord and electrophysiology, Kimura and Higashijima demonstrate that engrailed positive interneurons referred to as spinal V1 interneurons provide indeed in-phase inhibition onto motor neurons that is critical for setting the locomotor frequency. In the absence of spinal V1 interneurons, the period for one cycle is longer and the locomotor frequency is reduced.

This study is a beautiful demonstration of the similarity of organisation of the spinal CPGs across species, and the role of spinal V1 interneurons in providing in-phase inhibition to regulate the cycle frequency - with a major effect to reduce the cycle frequency during fast locomotor frequency (a factor 2 on bursting frequency in the fast regime, and only about 15% in the slow regime).

Furthermore, the authors have discovered a novel function for spinal V1 interneurons. They demonstrate that V1 interneurons provide during fast fictive locomotion a critical inhibition onto slow motor neurons that leads to their specific silencing in the fast regime.

The authors performed nice voltage clamp recordings at different holding potentials to isolate the excitatory and inhibitory components during the locomotor cycle - an approach pioneered in the spinal cord field by the McLean lab. Combined with genetic ablation of the spinal V1 interneurons, the authors show the contribution of the spinal V1 interneurons to the massive inhibition of slow motor neurons during the fast regime of fictive locomotion.

The authors show that this effect is mediated by a direct inhibition of slow motor neurons as well as an indirect effect onto slow V2a interneurons.

The authors perform double muscle cell recordings to show the impact of the premotor and motor effects of V1 onto the recruitment of slow and fast muscle fibers.

Minor comments:

1- It would be nice to show a representative video of the behavior of transgenic animals where spinal V1 interneurons were ablated to illustrate the reduction of tail beat frequency? The illustration of the effect on active locomotion would also potentially raise new questions for subsequent studies on possible effects on amplitude of swimming - not only frequency. A parameter one cannot probe quantitatively using ventral nerve root recordings.

2- In Fig 4e, the authors show very nice illustration of the change in inhibition onto motor neurons when spinal V1 interneurons are ablated. It would be interesting to use these graphs to quantitatively compare the level of excitation received by motor neurons in this condition.

Reviewer #3:

Remarks to the Author:

This paper investigates the function of V1 inhibitory interneurons in the context of the spinal cord central pattern generator circuit in zebrafish. The authors use evoked swimming (ES) in young (3 day old) zebrafish as a model. These ES swims are interesting, because they contain a transition from fast swimming frequencies to slow swimming frequencies and this has been associated with shifts in the active populations at the level of excitatory interneurons but also motor neurons and muscle types. Using a transgenic line that specifically labels spinal V1 neurons, they make electrophysiological recordings and show that this group has slow and fast swim-specific populations that both fire in phase with the motor output. Genetic ablation experiments show that these neurons (as was expected from other models) are responsible for allowing high frequency swimming, by shortening the contraction duration. In addition to this, they make the novel finding that these neurons are an essential component of the slow/fast circuit switch. They provide strong in phase inhibition to both slow and fast motor neurons during fast swimming, which prevents slow motor neurons from firing, and also act via excitatory V2 interneurons to reduce excitation of slow motor neurons during fast swimming. These elegant experiments add a very important set of components to the model of how this switch in gait patterns occurs, and will be extremely interesting to anyone working in the area of locomotion, spinal cord and motor circuits generally.

The experiments are well executed, the data is very clearly described and presented, and the results are clear, and support well the conclusions of the study. In summary, it is an excellent paper, which I would be happy to see published in its current form.

I have some very minor comments, which the authors may be interested take into account:

a) I found the use of the term 'instructive' a little confusing. I have usually heard this term used in terms of neural signals that 'instruct' a process of learning/plasticity. Perhaps there is another use of the term I am not familiar with, but I would suggest using a different term to avoid confusion.

b) It appears to me that most of the ablation results are explained in terms of a role for the 'fast' V1 neurons, and there is relatively little evidence in this data for a role of the 'slow' V1 neurons in the behaviors being studied. I wonder if the authors agree with this, and if they want to discuss more explicitly the function of the slow neurons?

c) The method for defining hybrid vs fast/slow neurons is not clearly described. Also some of the plots that consist of connected pairs of measurements (eg. Figure 1h) might be easier to interpret as a 2D scatter plot.

d) Tubocurarine is misspelled in several places

e) When describing the enhancers used to drive expression, the authors should be more specific about the sequences used (e.g. give 5' and 3' sequences).

Point-by-point responses to the reviewers:

Reviewer #1:

Reviewer's comment:

- 1) The interpretation of the data is made very difficult because of the change in the frequency after ablation of V1 INs. The following issues need to be addressed:
- i) Why do fast neurons become recruited at slow frequencies in V1 ablated fish?
 - ii) Is the lack of suppression of slow neurons activity simply due to the fact that the swimming does not reach frequencies as high as in control?
 - iii) Is the change seen only due to absence of V1 INs or to an unmasking of the effect of other neurons?
 - iv) Does ablation of V1 affect escape behavior?

Our response:

i) As the reviewer pointed out, the interpretation of the data is difficult because of the change in the frequency after ablation of V1 neurons. V1 neurons have two functions: frequency regulation and suppressing the activities of slow-component neurons during fast/strong movements. Unfortunately, these two are not separable. Although En1-DTA fish swim at slow frequencies (27-28 Hz) all the time, their swimming during the ES swim period (150 ms after electrical stimulation) is qualitatively different from the subsequent steady-state swim. This is apparent in the movie provided (Supplementary movie 1). Supplementary movie 1 presented in the original manuscript was a large file, and the reviewer may have had problems viewing it. (Reviewer #2 seemed to have the same problem.) Therefore, we changed the format of the movie to make the file smaller, and we hope the reviewer can now view it. As the movie shows, fish perform very strong movements during the ES swim period; the movement was clearly different from the later-phase of steady-state swim. Presumably, a high-level of supra-spinal inputs is provided to the spinal cord during the ES swim period. A high level of excitation is also apparent in the voltage clamp recording from fast-type MNs (Supplementary Fig. 7 in the revised manuscript). The figure shows that fast-type MNs both in control and En1-DTA fish receive very strong excitatory currents whose amplitude reaches tens of hundred pA (Supplementary Figs. 7a and b). These strong excitatory currents make fast-type MNs fire during the ES swim period even though swimming frequency is slow. Unlike in control fish, swimming frequency does not reflect the strength of the movement (or the level of excitation).

To make these points clearer we added the following new sentences to the revised manuscript:

"As will be described in the following sections, MNs did indeed receive very strong excitation during this period."

"In En1-DTA fish, the duration of each bending, including the escape bend, was extremely prolonged."

"In control fish, strong swim appears as a form of fast (high frequency) swim. In the case of En1-DTA fish, the frequency of the swimming (nearly constant at around 27-28 Hz) does not reflect the strength of the movement."

We think these modifications, together with an updated Supplementary movie 1, clarify these points.

ii) We do not think so. The lack of suppression is due to the absence of strong in-phase inhibition coming from V1 neurons (Figs. 4b and d). It is unrelated to the swimming frequency, as described above. We think that the added sentences described above make this point clear.

iii) We think the change seen is due to the absence of V1 given that (1) the timing of in-phase inhibition perfectly coincides with the timing of V1 neurons and that (2) there are direct inhibitory synaptic connections from V1 neurons to slow-type motor neurons (Supplementary Fig. 8). Although we (and anybody for that matter) cannot completely rule out the possibility that an unmasking of the effect of other neurons would, to some extent, contribute to the phenomenon, we think the evidence we provide is strong enough to draw the conclusion.

iv) As was apparent in Supplementary movie 1, escape behaviors are also affected. The duration of the escape bend is prolonged, which delays the start of the counter bend. We also examined sound/vibration evoked escapes at 5 dpf, and obtained essentially the same results. In summary, escape behaviors are affected in En1-DTA fish with the duration of escape bend prolonged. This is indicated in the text as follows:

"In En1-DTA fish, the duration of each bending, including the escape bend, was extremely prolonged."

Reviewer's comment:

2) Is the suppression simply due to the use of electrical stimulation that enables recruitment of fast (maybe escape related) neurons or is it the result of an increase in swimming frequency? This could be addressed by examining spontaneous swimming with increased frequency to see if during a spontaneous increase in frequency there is also a suppression of the activity of slow swimming neurons.

Our response:

The suppression is not simply due to the use of electrical stimulation. The phenomenon

also occurs when swimming frequency is high during Non-ES swim. This was illustrated in muscle-cell recordings in the previous manuscript (Fig. 6). A corresponding slow-type-MN recording (suppression of spiking activity at high-frequency swimming during Non-ES swim) is now presented in Supplementary Figure 12 in the revised manuscript.

Reviewer's comment:

3) Related to the previous point, the circuit mechanism proposed is reminiscent of what has been shown in adult zebrafish. In a study by Song et al (Current Biology, 2015), it was shown that escape momentarily suppresses swimming via activation of an inhibitory class of interneuron possibly V1. This study should be cited and discussed in the context of primacy of escape over swimming.

Our response:

In the revised manuscript, the paper (Song et al., Current Biology, 2015) is cited in the Introduction. Because the escape behavior is not the focus of the current study, we avoided a detailed discussion of this matter.

Reviewer's comment:

4) How does the ablation of V1 neurons in the hindbrain contribute to the observed effects? The authors should at least acknowledge this in the discussion.

Our response:

In response to this comment, we performed a new experiment in which a different *hox:Cre* (Tg[*hoxa9a-3'*enhancer:Cre]) was used for the ablation of V1 neurons. In the Tg[*hoxa9a-3'*enhancer:Cre] line, Cre is only expressed in the spinal cord, not in the hindbrain, and, consequently, V1 neurons in the hindbrain remained almost intact (Supplementary Fig. 6a). The use of the new Cre line produced the same results, indicating that the phenotypes were primarily caused by the absence of spinal V1 neurons. The results are presented in Supplementary Figure 6b.

The reason that we did not mainly use the Tg[*hoxa9a-3'*enhancer:Cre] line in the current study was because Cre is expressed in the trunk muscle cells in this line. (Note that RFP expression in slow muscle cells in the middle region of the trunk was absent in the compound transgenic fish; Supplementary Fig. 6a.) This resulted in the ablation of the corresponding slow muscle cells, and we were afraid that this would make recordings from slow-muscle cells (Fig. 6) more difficult.

Reviewer's comment:

5) The paper would gain in strength if the authors could provide some connectivity data supporting the schematic in Figure 7. Do fast V1 IN connect to all slow neurons or only to some? Do slow V1 INs connect or not to fast neurons? Such results could strengthen the authors' conclusions. Without such data the proposed role of V1 remains speculative.

Our response:

Based on this comment, we performed paired recordings. We showed direct connections between (1) fast-type V1 neurons and slow-type MNs and (2) fast-type V1 neurons and fast-type MNs. The results are shown in Supplementary Figures 8 and 9, respectively. The reviewer also asked about possible connections from slow V1 neurons to fast neurons. Unfortunately, this is not a point of the present study. In the schematic shown in Figure 7, such connections are not proposed. For this reason, we did not attempt paired recordings between slow-type V1 neurons and fast-type neurons.

Reviewer's comment:

6) To assess the pattern of activity of the different neurons during fast vs slow swimming, it would be better to show current clamp data showing the change in membrane potential of fast vs slow V1, MNs and V2a's during swimming in control and in ablated fish (MNs and V2a's). The data as shown only inform about the firing of neurons. These neurons could still be receiving subthreshold rhythmic excitation; therefore, it is necessary to provide current clamp data.

Our response:

The reviewer asked us to provide current clamp data to show the change in membrane potential. As for V1 neurons, we think that providing such data is beyond the scope of the present study. The focus of the manuscript is the functions of V1 neurons; the study does not aim to address how the firings of V1 neurons are regulated. As for MNs, the manuscript has already presented loose-patch recordings (Fig. 3) and voltage-clamp recordings (Fig. 4 and Supplementary Fig. 7). The former shows the firing patterns of the neurons through the least invasive method and the latter shows synaptic currents with excitatory and inhibitory currents being separated. Thus, the combination of the two provides richer information than current clamp recordings would, in which both excitatory and inhibitory inputs appear as a depolarizing response under our recording conditions. For this reason, we do not think that providing additional current clamp data is necessary for MNs. It should be noted that, though it was unintentional, current clamp recordings of slow-type and fast-type MNs in control animals are shown in Supplementary Figures 8 and 9, respectively. As for V2a neurons, current clamp recordings are shown in Supplementary Figure 10 in the revised manuscript.

Reviewer's comment:

7) Figure 2: It is confusing to use both fast vs slow swim and strong vs weak swim. The main effect of V1 ablation is that there is a decrease in the frequency of stimulation-induced locomotion with little effect on the frequency of spontaneous swimming. The lack of suppression of the activity of slow/weak neurons could be just the reflection of fact that escape or fast swimming is compromised.

Our response:

As noted earlier (reply to comment #1), En1-DTA fish perform strong swimming during the ES swim period. Because their swimming frequency is not high, we cannot use the term "fast" for their strong movements. As mentioned earlier, swimming frequency does not reflect the strength of the movements in En1-DTA fish. We think that the term "strong" is the best way to describe their swimming during the ES swim period. Similarly, because swimming frequency is constant regardless of the strength of movements, we need to use the term "weak" instead of slow during the period when fish perform steady-state weak swim. (In control fish, such swimming appears as slow swim.) We think the addition of the sentences in the revised manuscript (see our reply to comment #1) clears up the confusion.

Reviewer's comment:

8) Figure 3: Please provide a comparison of the minimum recruitment frequency of the fast vs slow MNs in control and in V1 ablated fish. Again, why do fast MNs become recruited at slow swimming frequency? According to the available data in larval zebrafish, fast neurons should not be recruited at frequencies below 30 Hz. Please clarify. Please show the distribution (box and whisker) for the right graph in Figure 3H (En1-DTA, slow MN).

Our response:

As mentioned earlier, the swimming frequency of En1-DTA fish is always low, regardless of the excitation level. During the ES swim period, fast MNs receive very strong excitatory inputs (Supplementary Fig. 7). These strong excitatory inputs make fast MNs fire during the ES swim period. Due to the absence of in-phase inhibition from V1 neurons, the cycle period gets prolonged, resulting in low-frequency swimming. As noted earlier, swimming frequency correlates to the strength of the movement in control fish. In En1-DTA fish, however, it can no longer be used to estimate the strength of the movement. In this sense, the minimum recruitment frequency does not have significant meaning in En1-DTA fish. (Swimming frequency is

almost constant regardless of the strength of the movements.) Because of this reason, we do not think that showing such data would provide the readers with useful information.

Reviewer's comment:

9) Figure 9: The fact that slow MNs are receiving rhythmic on-cycle excitatory current and tonic inhibition. This shows that the excitatory interneurons driving locomotion are still active and provide rhythmic excitatory drive to slow MNs. These results also indicate that the effect of suppression of slow swimming is mainly occurring at the level of MNs and not at the level of INs (V2a's). This is similar to what was shown in adult zebrafish by Song et al (Current Biology, 2015).

The inhibition of firing of V2a INs shown in Figure 5 could be the results of electrical coupling that allows for the propagation of inhibition from MNs retrogradely to V2a INs (see Song et al., Nature, 2016). This should be at least mentioned in the discussion.

Our response:

In frog tadpoles, V1 neurons were shown to make direct inhibitory connections onto V2a-corresponding neurons (dINs in *Xenopus*; Li et al., 2004). Given this, we predict that there are direct inhibitory connections between V1 neurons and V2a neurons, and we expect such connections contribute to the inhibition of the firing of V2a neurons shown in Figure 5. Nonetheless, this does not rule out the possible inhibition coming from MNs via gap junctions, as the reviewer points out. We cite Song et al. (Nature, 2016) and mention possible inhibitory effects coming from MNs by electrical coupling between MNs and V2a neurons in the revised manuscript.

Reviewer's comment:

10) Figure 6: Why do slow muscle fibers receive a strong excitatory current during fast swimming? Shouldn't they be quiescent? If the activity of slow MNs is suppressed during fast swim, there shouldn't be any excitatory current in slow muscle fibers. The recording shows that slow muscle fibers receive excitation during fast locomotion perhaps because of the slow muscles are also innervated by fast MNs. I think it is more appropriate to compare the current (Figure 6e) only in the portion of the recordings where the fast muscle fibers are quiescent.

It seems that slow muscle fibers are also innervated by fast MNs and as such their activity is not suppressed during strong/fast locomotion. The existence of inward current in slow muscle fibers during activity of fast MNs in the absence of slow MN activity should be mentioned and discussed.

Our response:

We think this is a misunderstanding on the part of the reviewer. Slow muscle fibers receive a very small excitatory current during fast swimming in control fish (Fig. 6).

Reviewer's comment:

11) In both the introduction and discussion, the authors should mention/discuss work from juvenile/adult zebrafish on the recruitment of MNs and V2a INs showing that there is no suppression of the activity of slow neurons during fast swimming and that this suppression only occurs during escape.

Our response:

Compared to larval fish, the swimming frequency of adult fish is generally low due to their larger body size. Because of this, suppression of slow-type neurons may only occur in a very-high frequency range for swimming in adult fish. Indeed, the previous studies in yellow tail (Tsukamoto, 1984) and bluegill sunfish (Jayne and Lauder) show that the suppression of slow muscle activity is seen during very high-speed swimming in adult fish. Previous studies in juvenile/adult zebrafish did not detect the suppression of slow MNs and V2a neurons (Ampatzis et al., 2013; 2014), but the maximum frequency examined in these studies was around 25 Hz. This, however, is not the maximum swimming frequency that adult zebrafish can achieve. In cases of emergency, adult fish can swim at a swimming frequency of over 30 Hz. In fact, the maximum frequency can reach around 40 Hz (Drs. Matsuda, Miyasaka and Yoshihara, personal communication). This raises the possibility that the suppression of slow MNs could occur during this type of super-fast swimming in adult zebrafish. Given this situation, it is potentially inaccurate to write that "there is no suppression of the activity of slow neurons during fast swimming and that this suppression only occurs during escape." Because of this reason, we did not discuss this point.

Reviewer #2:

This reviewer has only minor comments.

Reviewer's comment:

1- It would be nice to show a representative video of the behavior of transgenic animals where spinal V1 interneurons were ablated to illustrate the reduction of tail beat frequency ? The illustration of the effect on active locomotion would also potentially raise new questions for subsequent studies on possible effects on amplitude of swimming

- not only frequency. A parameter one cannot probe quantitatively using ventral nerve root recordings.

Our response:

The video was presented in the previous manuscript. The reviewer may have had difficulty in viewing it on the web because the file was large. In the revised manuscript, we changed the format of the movie to make the file smaller, and we hope this will solve the problem.

2- In Fig 4e, the authors show very nice illustration of the change in inhibition onto motor neurons when spinal V1 interneurons are ablated. It would be interesting to use these graphs to quantitatively compare the level of excitation received by motor neurons in this condition.

Our response:

In Figure 4e, the level of excitation received by motor neurons is indicated in the same graph.

Reviewer #3:

This reviewer has only minor comments.

Reviewer's comment:

a) I found the use of the term 'instructive' a little confusing. I have usually heard this term used in terms of neural signals that 'instruct' a process of learning/plasticity. Perhaps there is another use of the term I am not familiar with, but I would suggest using a different term to avoid confusion.

Our response:

In the original manuscript, the term "instructive" was used in three places: the title, introduction, and discussion. In each instance, we modified the word (or sentence). This led to the alteration of the title.

Reviewer's comment:

b) It appears to me that most of the ablation results are explained in terms of a role for the 'fast' V1 neurons, and there is relatively little evidence in this data for a role of the 'slow' V1 neurons in the behaviors being studied. I wonder if the authors agree with this, and if they want to discuss more explicitly the function of the slow neurons?

Our response:

As the reviewer points out, the present manuscript mostly focuses on the function of fast-type V1 neurons. Detailed functional analyses of slow-type V1 neurons will be reported in future studies. We mention this in the revised manuscript.

Reviewer's comment:

c) The method for defining hybrid vs fast/slow neurons is not clearly described. Also some of the plots that consist of connected pairs of measurements (eg. Figure 1h) might be easier to interpret as a 2D scatter plot.

Our response:

The definition of fast-, slow-, and hybrid-type neurons was indicated in the legend of Figure 1. In the revised manuscript, the definition of hybrid-type neurons is also indicated in the legend of Supplementary Figure 11.

As for 2D scatter plots, we attempted to create figures, but were not able to create sufficiently good images. Therefore, we did not employ 2D scatter plots.

Reviewer's comment:

d) Tubocurarine is misspelled in several places

Our response:

We corrected the misspellings.

Reviewer's comment:

e) When describing the enhancers used to drive expression, the authors should be more specific about the sequences used (e.g. give 5' and 3' sequences).

Our response:

In the revised manuscript, 5' and 3' sequences for the enhancer used are indicated.

Reviewers' Comments:

Reviewer #1:

Remarks to the Author:

This interesting body of work is greatly improved with the current revision. The authors should be commended for their thorough revision of this paper, which will be of interest to the broad readership of this Journal.

Reviewer #2:

Remarks to the Author:

From my prospective the authors satisfactorily answered the technical points on the data themselves raised by the reviewers. Some of the ambiguities raised in the first version of the manuscript are solved with a more detailed explanation provided by the authors. I only have below minor suggestions regarding the information discussed / emphasized by the authors (see below).

1- The data of Kimura and Higshijima are very compelling especially with data requested by Reviewer 1 :

A- the nice addition of the Hox9a line enabling restriction of DTA to the spinal cord only (despite minor muscle expression) confirming the results where DTA is expressed as well in caudal hindbrain,

B- the addition of double paired patch recordings between fast V1 premotor interneurons and either slow or fast motor neurons to validate the proposed connectivity model.

=> Here, It would be interesting to specify, as the probability observed are quite low (2 out of 12 and 2 out of 7), whether the authors used anatomy of V1 projections onto somas of MNs to guide their chance to find connections?

2- There are two points that I find important and that the authors currently do not emphasise much in the abstract and discussion of the revised manuscript:

A) From the behavior analysis point of view, it is clear indeed that in zebrafish larva during escapes (and most likely also during fast swimming referred to as Burst Swimming), tail beat frequency correlates with strength of the movement and slow swims (not struggles) in control zebrafish larva.

What is remarkable in the data presented by Kimura and Higashijima is the fact that in V1-ablated fish, these kinematic variables (amplitude and tail beat frequency) are now decoupled during fast swimming.

=> It would be a nice addition to emphasize the precise roles of V1 in title, and add this effect in the abstract.

B) While most of us in the field has been mainly focused up to now at understanding the recruitment of spinal neurons as a function of locomotor frequency (usually reflecting locomotor speed), there is evidence from this manuscript and the citable manuscript of Minoru Koyama and Avinash Pujala in BioRxiv (URL: <https://www.biorxiv.org/content/10.1101/425587v1>) that under normal conditions during struggles or when spinal V1 interneurons are silenced, fast and large amplitude movements can

be decoupled. This leads Kimura and Higashijima to employ the terms strong and weak and Koyama and Pujala crude and refined.

1. Ideally, we should all converge on a terminology where we only refer to vigor (often measuring amplitude) and speed (often measuring frequency) and this manuscript could be an important step towards adopting the same vocabulary: what do you think ?

2. It would be worth citing Koyama and Pujala's manuscript in BioRxiv and discussing in the manuscript the distinction that should be made between the coupling of speed / tail beat frequency and amplitude of movement / strength in different contexts.

3- Adding to the discussion:

I think the discussion of the results described here on the role of V1 during fast swimming in light of the observation of the effect observed by Song et al during escapes in Current Biology 2015 would be a nice addition.

4- Overall the study raises interesting follow-up studies out of the scope of this study but that the authors may consider briefly discussing as well (I understand if these requests cannot be implemented as the discussion is quite substantial):

a- the question for future studies on the contribution of slow spinal V1 interneurons raised by Reviewer 1

b- the mysterious role (if any) of hindbrain V1 interneurons found highly recruited during locomotion (Severi et al Scientific Reports 2018).

Note here - Confirming the results from the authors, we had observed using Botulinum toxin a similar reduction of speed associated with larger amplitude movements during escapes in Tg(eng1b:gal4;UAS:BotxLCB-GFP) (unpublished data). I find it remarkable that the same effects can be induced by solely silencing spinal V1 interneurons and wonder what hindbrain V1 interneurons recruited during locomotion may be contributing to.

To sum up, my requests are only ****minor**** revisions of the current manuscript that would be highly appreciated for publication.

Reviewer #3:

Remarks to the Author:

The authors have satisfactorily addressed my previous comments in their response and in revisions to the manuscript. I recommend that it be accepted.

Point-by-point responses to the reviewers:

Reviewer #1 (Remarks to the Author):

This interesting body of work is greatly improved with the current revision. The authors should be commended for their thorough revision of this paper, which will be of interest to the broad readership of this Journal.

Reviewer #2

Reviewer's comment:

1- The data of Kimura and Higshijima are very compelling especially with data requested by Reviewer 1 :

A- the nice addition of the Hox9a line enabling restriction of DTA to the spinal cord only (despite minor muscle expression) confirming the results where DTA is expressed as well in caudal hindbrain,

B- the addition of double paired patch recordings between fast V1 premotor interneurons and either slow or fast motor neurons to validate the proposed connectivity model.

=> Here, It would be interesting to specify, as the probability observed are quite low (2 out of 12 and 2 out of 7), whether the authors used anatomy of V1 projections onto somas of MNs to guide their chance to find connections?

Our response:

We randomly selected neurons for each pair. Based on the reviewer's comment, we added the word "randomly" to the corresponding sentence in the methods section.

2- There are two points that I find important and that the authors currently do not emphasise much in the abstract and discussion of the revised manuscript:

A) From the behavior analysis point of view, it is clear indeed that in zebrafish larva during escapes (and most likely also during fast swimming referred to as Burst Swimming), tail beat frequency correlates with strength of the movement and slow swims (not struggles) in control zebrafish larva.

What is remarkable in the data presented by Kimura and Higashijima is the fact that in V1-ablated fish, these kinematic variables (amplitude and tail beat frequency) are now decoupled during fast swimming.

=> It would be a nice addition to emphasize the precise roles of V1 in title, and add this effect in the abstract.

Our response:

Taking the reviewer's comment into account, we changed the title to "Regulation of locomotor speed and selection of active sets of neurons by V1 neurons." We did not change the abstract because of the word limit. Instead, we added a discussion about the issue of decoupling between speed and strength of the movement (in response to the reviewer's comment 2B2, see below).

B) While most of us in the field has been mainly focused up to now at understanding the recruitment of spinal neurons as a function of locomotor frequency (usually reflecting locomotor speed), there is evidence from this manuscript and the citable manuscript of Minoru Koyama and Avinash Pujala in BioRxiv (URL:

<https://www.biorxiv.org/content/10.1101/425587v1>) that under normal conditions during struggles or when spinal V1 interneurons are silenced, fast and large amplitude movements can be decoupled. This leads Kimura and Higashijima to employ the terms strong and weak and Koyama and Pujala crude and refined.

1. Ideally, we should all converge on a terminology where we only refer to vigor (often measuring amplitude) and speed (often measuring frequency) and this manuscript could be an important step towards adopting the same vocabulary: what do you think ?

Our response:

We agree with the reviewer: vigor/vigorous is a good term to describe large amplitude movements. However, the words strong and weak are used as a pair in the present manuscript. Typical antonyms of vigorous (e.g., feeble) do not seem appropriate to describe the steady-state swim in En1-DTA fish. In this sense, we think the terms strong and weak are appropriate in the present manuscript.

2. It would be worth citing Koyama and Pujala's manuscript in BioRxiv and discussing in the manuscript the distinction that should be made between the coupling of speed / tail beat frequency and amplitude of movement / strength in different contexts.

Our response:

In response to the reviewer's comment, we added the following sentence to the

discussion section:

Defects in frequency regulation during the fast-swim period resulted in low-frequency tail beat with very large body bend (Supplementary Movie 1; decoupling of tail-beat frequency and the amplitude of body bend are also seen during struggling [Pujala and Koyama, 2019]).

3- Adding to the discussion:

I think the discussion of the results described here on the role of V1 during fast swimming in light of the observation of the effect observed by Song et al during escapes in Current Biology 2015 would be a nice addition.

Our response:

We did cite the paper (Song et al., Current Biology, 2015) in the introduction in the first revision of the paper. Because the escape behavior is not the focus of our study, we avoided a detailed discussion of this matter.

4- Overall the study raises interesting follow-up studies out of the scope of this study but that the authors may consider briefly discussing as well (I understand if these requests cannot be implemented as the discussion is quite substantial):

a- the question for future studies on the contribution of slow spinal V1 interneurons raised by Reviewer 1

Our response:

This was a comment from Reviewer 3, instead of Reviewer 1. We did address this issue in the first revision of the manuscript.

b- the mysterious role (if any) of hindbrain V1 interneurons found highly recruited during locomotion (Severi et al Scientific Reports 2018).

Note here - Confirming the results from the authors, we had observed using Botulinum toxin a similar reduction of speed associated with larger amplitude movements during escapes in Tg(eng1b:gal4;UAS:BotxLCB-GFP) (unpublished data). I find it remarkable that the same effects can be induced by solely silencing spinal V1 interneurons and wonder what hindbrain V1 interneurons recruited during locomotion may be

contributing to.

Our response:

The role of hindbrain V1 neurons in locomotion is very interesting. However, the focus of our manuscript is spinal V1 neurons, and thus, we avoided a detailed discussion of hindbrain V1 neurons.

To sum up, my requests are only ****minor**** revisions of the current manuscript that would be highly appreciated for publication.

Reviewer #3 (Remarks to the Author):

The authors have satisfactorily addressed my previous comments in their response and in revisions to the manuscript. I recommend that it be accepted.